# One Health surveillance of colistin-resistant Enterobacterales in Belgium and the Netherlands between 2017 and 2019

Sien De Koster[1], Basil Britto Xavier[1,2,3,4], Christine Lammens[1], Natascha Perales Selva[5], Stefanie van Kleef-van Koeveringe[5], Samuel Coenen[1], Youri Glupczynski[1], Isabel Leroux-Roels[6], Wouter Dhaeze[7], Christian J. P. A. Hoebe[8,9,10], Jeroen Dewulf[11], Arjan Stegeman[12], Marjolein Kluytmans-Van den Bergh[13,14,15], Jan Kluytmans[13,15,16], Herman Goossens[1]*, i-4-1-Health Study Group[¶]

1 Laboratory of Medical Microbiology, Vaccine and Infectious Disease Institute, University of Antwerp, Antwerp, Belgium, 2 Department of Clinical Sciences, Institute of Tropical Medicine, HIV/STI Unit, Antwerp, Belgium, 3 Hospital Outbreak Support Team-HOST, ZNA Middelheim, Antwerp, Belgium, 4 Hospital Outbreak Support Team-HOST, GZA Ziekenhuizen, Wilrijk, Belgium, 5 Laboratory of Medical Microbiology, Antwerp University Hospital, Antwerp, Belgium, 6 Laboratory of Medical Microbiology and Infection Control Department, Ghent University Hospital, Ghent, Belgium, 7 Department Care, Brussels, Belgium, 8 Department of Social Medicine, Maastricht University, Maastricht, the Netherlands, 9 Department of Medical Microbiology, Infectious Diseases and Infection Prevention, Maastricht University Medical Center+, Maastricht, The Netherlands, 10 Living Lab Public Health, Public Health Service South Limburg, Heerlen, the Netherlands, 11 Faculty of Veterinary Medicine, Department of Internal Medicine, Reproduction and Population Medicine, Veterinary Epidemiology Unit, Ghent University, Merelbeke, Belgium, 12 Faculty of Veterinary Medicine, Department of Population Health Sciences, Utrecht University, Utrecht, The Netherlands, 13 Department of Infection Control, Amphia Hospital, Breda, The Netherlands, 14 Amphia Academy Infectious Disease Foundation, Amphia Hospital, Breda, The Netherlands, 15 Department of Medical Microbiology, University Medical Center Utrecht, Utrecht University, Utrecht, The Netherlands, 16 Microvida Laboratory for Microbiology, Amphia Hospital, Breda, The Netherlands

¶ Membership of the i-4-1-Health Study Group is provided in the Acknowledgments
* herman.goossens@uza.be

**Data Availability Statement:** All relevant data are within the manuscript and its Supporting Information files. The sequences were submitted in NCBI under BioProject PRJNA927131.

## Abstract

### Background

Colistin serves as the last line of defense against multidrug resistant Gram-negative bacterial infections in both human and veterinary medicine. This study aimed to investigate the occurrence and spread of colistin-resistant Enterobacterales (ColR-E) using a One Health approach in Belgium and in the Netherlands.

### Methods

In a transnational research project, a total of 998 hospitalized patients, 1430 long-term care facility (LTCF) residents, 947 children attending day care centres, 1597 pigs and 1691 broilers were sampled for the presence of ColR-E in 2017 and 2018, followed by a second round twelve months later for hospitalized patients and animals. Colistin treatment incidence in livestock farms was used to determine the association between colistin use and resistance. Selective cultures and colistin minimum inhibitory concentrations (MIC) were employed to identify ColR-E. A combination of short-read and long-read sequencing was utilized to investigate the molecular characteristics of 562 colistin-resistant isolates. Core genome

**Funding:** The i-4-1-Health project was financed by the Interreg V Flanders-The Netherlands program, the cross-border cooperation program with financial support from the European Regional Development Fund (ERDF) (0215). Additional financial support was received from the Dutch Ministry of Health, Welfare and Sport (325911), the Dutch Ministry of Economic Affairs (DGNR-RRE/14191181), the Province of Noord-Brabant (PROJ-00715/PROJ-01018/PROJ-00758), the Belgian Department of Agriculture and Fisheries (no reference), the Province of Antwerp (1564470690117/1564470610014) and the Province of East-Flanders (E01/subsidie/VLNL/i-4-1-Health). The authors are free to publish the results from the project without interference from the funding bodies. FecalSwabs and tryptic soy broths were provided by Copan. The authors were free to publish the results from the project without interference by Copan.

**Competing interests:** The authors have declared that no competing interests exist.

multi-locus sequence typing (cgMLST) was applied to examine potential transmission events.

## Results

The presence of ColR-E was observed in all One Health sectors. In Dutch hospitalized patients, ColR-E proportions (11.3 and 11.8% in both measurements) were higher than in Belgian patients (4.4 and 7.9% in both measurements), while the occurrence of ColR-E in Belgian LTCF residents (10.2%) and children in day care centres (17.6%) was higher than in their Dutch counterparts (5.6% and 12.8%, respectively). Colistin use in pig farms was associated with the occurrence of colistin resistance. The percentage of pigs carrying ColR-E was 21.8 and 23.3% in Belgium and 14.6% and 8.9% in the Netherlands during both measurements. The proportion of broilers carrying ColR-E in the Netherlands (5.3 and 1.5%) was higher compared to Belgium (1.5 and 0.7%) in both measurements. *mcr*-harboring *E. coli* were detected in 17.4% (31/178) of the screened pigs from 7 Belgian pig farms. Concurrently, four human-related *Enterobacter* spp. isolates harbored *mcr-9.1* and *mcr-10* genes. The majority of colistin-resistant isolates (419/473, 88.6% *E. coli*; 126/166, 75.9% *Klebsiella* spp.; 50/75, 66.7% *Enterobacter* spp.) were susceptible to the critically important antibiotics (extended-spectrum cephalosporins, fluoroquinolones, carbapenems and aminoglycosides). Chromosomal colistin resistance mutations have been identified in globally prevalent high-risk clonal lineages, including *E. coli* ST131 (n = 17) and ST1193 (n = 4). Clonally related isolates were detected in different patients, healthy individuals and livestock animals of the same site suggesting local transmission. Clonal clustering of *E. coli* ST10 and *K. pneumoniae* ST45 was identified in different sites from both countries suggesting that these clones have the potential to spread colistin resistance through the human population or were acquired by exposure to a common (food) source. In pig farms, the continuous circulation of related isolates was observed over time. Inter-host transmission between humans and livestock animals was not detected.

## Conclusions

The findings of this study contribute to a broader understanding of ColR-E prevalence and the possible pathways of transmission, offering insights valuable to both academic research and public health policy development.

## Introduction

Colistin (polymyxin E) has been classified by the World Health Organization (WHO) as critically important for human medicine with the highest priority [1]. It is also recognized as an antibiotic of high importance in veterinary medicine by the World Organization for Animal Health (OIE) [2]. Colistin is administered orally in animals for the treatment of gastrointestinal infections and sepsis caused by Enterobacterales in intensive husbandry systems, mainly in swine and poultry [3–5]. In healthcare settings, colistin is a reserve antibiotic for multidrug-resistant (MDR) Gram-negative infections [1,4,6] and it is also used for the treatment of *P. aeruginosa* infections in cystic fibrosis patients, topical treatment of otitis externa or ophthalmic infections [4] and for selective decontamination in critically ill patients [7,8]. With the increasing number of hospital outbreaks with carbapenemase-producing Enterobacterales (mostly

*Klebsiella* species) and MDR non-fermentative Gram-negative bacteria (*Pseudomonas* and *Acinetobacter* species), colistin plays a key role for public health [3,9]. The rising incidence of MDR and colistin-resistant Gram-negative Enterobacterales among the human and animal populations has led to a lack of effective therapeutic approaches for these infections, resulting in suboptimal clinical outcomes [4].

The emergence of colistin resistance is primarily due to alterations in lipopolysaccharide (LPS), the primary target site for this antibiotic [10,11]. Such modification can result from chromosomal mutations that cause overexpression of the *pmrHFIJKLM* operon, *pmrCAB* operon and the *pmrE* gene, as well as the presence of plasmid-mediated mobile colistin resistance (*mcr*) genes. As many as eleven plasmid replicon types, including IncI2, IncX4, IncP, IncX, and IncFII, have been linked to the transmission of colistin-resistance genes [12,13]. Furthermore, these plasmids exhibit a high degree of stability [14]. Colistin resistance genes have been isolated from poultry, pigs, cattle, animal-derived food products and human isolates [15].

In the context of the global dissemination of colistin resistance, key contributing factors are the international trade of food animals and meat, as well as global mobility of colonized or infected individuals [16]. A meta-analysis has revealed that the primary reservoirs of *mcr*-harboring *E. coli* were found in chickens and pigs with estimated global prevalences of 15.8% and 14.9%, respectively. Lower prevalences of plasmid-mediated colistin resistance were observed in *E. coli* isolates from healthy human populations (7.4%) and clinical samples (4.2%) [13]. Evidence of clonal transmission within the livestock sectors and into the meat sectors exists [17,18]. *mcr* genes were also detected in wastewater, rivers and seawater [14,19,20] and in dog feces and flies [14]. This highlights the importance of an integrated, multisectoral approach that fits within the concept of One Health-*i.e.* across human, animal and environmental health. However, currently, surveillance systems in livestock and humans are heterogeneous in Europe [21]. In 2014, European monitoring for colistin resistance in *Salmonella* and indicator microorganism *E. coli* in animals became mandatory (Regulation 2013/652/EU) [3]. In contrast, surveillance of colistin resistance in Gram-negative clinical isolates from humans is not yet implemented in Europe. Consequently, it is crucial to monitor the presence and transmission of antibiotic resistance in key reservoirs, such as humans, chickens and pigs in order to effectively combat the emergence and spread of colistin-resistant bacteria and colistin resistance genes. Current data from global studies describing the circulation of colistin-resistant bacteria among humans, animals, food and the environment is scarce.

Utilizing a One Health approach with harmonized and comparable methodology, our study examines the prevalence and possible dissemination of colistin-resistant Enterobacterales (ColR-E) in hospital patients, long-term care facility (LTCF) residents and healthy children in day care centres, as well as in broilers and pigs on farms in Belgium and the Netherlands. We also aimed to elucidate the molecular basis of colistin resistance in different human healthcare settings and in livestock farming environments.

## Materials and methods

### Setting, study period and sample/strain collection

This study is part of the i-4-1-Health Interreg project, a One Health project on the prevalence and spread of antimicrobial resistance in the human and veterinary domain in the Dutch-Belgian cross-border region. An analysis of 6591 fecal, perianal or gastrointestinal stoma samples was conducted. These samples were obtained from hospitalized patients (n = 998), LTCF residents (n = 1430), children attending day care centres (n = 947), pigs (n = 1597) and broilers (n = 1619) across Flanders, Belgium and the South of the Netherlands.

The recruitment and collection period spanned from 25 September 2017 to 22 February 2019. The samples originated from different sites: three hospitals (one from Belgium and two from the Netherlands), 30 LTCFs (thirteen from Belgium and seventeen from the Netherlands), 45 day care centres (seventeen from Belgium and 28 from the Netherlands), 31 multiplier pig farms (fifteen from Belgium and sixteen from the Netherlands) and 29 broiler farms (fifteen from Belgium and fourteen from the Netherlands) (S1 Table). Sites were included based on the location (Belgian-Dutch border region). Additionally, the farms were included based on the farm type (conventional broiler farms and multiplier pig farms) and relative level of antibiotic use which exceeded the average use compared to the national benchmark value in the respective countries. Farm characteristics and antibiotic use were described previously [22]. Screened patients were hospitalized in four different wards including at least one surgical unit and an internal medicine ward in each hospital. Screening for rectal carriage was performed on a single day every two weeks in a two months time period. In the farms, samples were collected in a stratified-random sampling design. The collection of 30 fecal samples per farm was aimed, evenly distributed over different units (broiler houses or rooms with weaned pigs) to take into account intra-farm variability.

Samples were collected cross-sectionally using a nylon-flocked swab with 2 mL Cary-Blair transport medium (FecalSwab™, Copan Italy, Brescia, Italy). Two rounds of repeated surveys, with a one-year interval between each measurement, were performed in hospitals and in livestock farms. A single survey was performed in long-term care facilities and in day care centres.

## Colistin use in livestock farms

Colistin use in the livestock farms was calculated from registration documents provided by national quality assurance organizations, the farmers or farm veterinarians (S2 Table). Antibiotic use was quantified as the treatment incidence per 100 days for pigs and per production round for broilers described by Caekebeke and colleagues (2020) [22].

## Isolation of colistin-resistant Enterobacterales and antibiotic susceptibility testing

Protocols followed for collection and culturing of specimens were similar in the two countries. Selective isolation of ColR-E was performed as previously described by Kluytmans-van den Bergh and colleagues [23]. Briefly, swabs were pre-enriched in a non-selective tryptic soy broth (Copan, Brescia, Italy) and directly cultured on blood agar as a positive control. After 18–24 hours of incubation at 35–37°C, enrichment broths were subcultured on selective agar (eosine methylene blue agar (Oxoid, Basingstoke, United Kingdom) supplemented with 3.5 mg/L colistin, 10 mg/L daptomycin and 5 mg/L amphoterin B (Sigma-Aldrich, Saint Louis, United States)). After 18–24 hours of incubation, isolates were identified with MALDI-TOF. All non-intrinsically resistant Enterobacterales species were subjected to broth microdilution (Micronaut MIC-Strip Colistin, Merlin Diagnostika GmbH, Bornheim, Germany) for colistin minimum inhibitory concentration (MIC) determination at the University of Antwerp. Reference strains *E. coli* ATCC25922 (colistin MIC: 0.25 mg/L), *P. aeruginosa* ATCC27853 (colistin MIC: 1 mg/L), *E. coli* NCTC 13846 (*mcr-1* positive, colistin MIC: 4 mg/L) and in-house *K. pneumoniae* 08400 (colistin MIC: 64 mg/L) were used as quality controls. Besides colistin, antibiotic susceptibility testing was performed with a distinct local panel for antibiotic susceptibility testing, by Amphia Hospital (Breda, the Netherlands) for the Dutch isolates (ampicillin, amoxicillin-clavulanic acid, piperacillin-tazobactam, cefoxitin, cefuroxime, ceftazidime, cefotaxime, ciprofloxacin, gentamicin, meropenem, trimethoprim-sulfamethoxazole 1:19, fosfomycin using VITEK 2® (N344), bioMérieux, Marcy l'Etoile, France) and by University of

Antwerp and Antwerp University Hospital for the Belgian isolates (ampicillin (10μg), amoxicillin–clavulanic acid (20/10μg), piperacillin–tazobactam (30/6μg), cefoxitin (30μg), cefuroxime (30μg), ceftriaxone (30μg), ceftazidime (10μg), ciprofloxacin (5μg), meropenem (10μg), amikacin (30μg), trimethoprim-sulfamethoxazole (1.25/23.75μg) and fosfomycin (200μg) using disk diffusion (Rosco, Taastrup, Denmark)) as described before [23]. The EUCAST breakpoints v12.0 (January 2022) were used for the interpretation of antibiotic susceptibility and resistance. Multidrug resistance is defined as resistance to at least one antimicrobial drug in three or more antibiotic classes [24].

## Short-and long-read sequencing of colistin-resistant Enterobacterales

Whole genome sequencing was performed on isolates identified as *Escherichia coli*, *Klebsiella* spp. and *Enterobacter* spp. Selection for sequencing was based on unique isolates exhibiting variations in susceptibility or resistance for at least one antibiotic class as well as two-fold (or larger) differences in colistin MIC, when multiple isolates were obtained from each individual or farms. This selection led to the whole genome sequencing of 562 colistin-resistant isolates. Additionally, 3 colistin-susceptible *E. coli* and 6 colistin-susceptible *K. pneumoniae* were sequenced and were used for comparison with resistant isolates within the study setting. Two colistin-resistant *K. pneumoniae* (1103990 and 1101433) and one colistin-susceptible *K. pneumoniae* (1101124) were selected for long-read sequencing on PacBio Sequel 1 (Pacific Biosciences, CA, USA). All other isolates were sequenced using the short-read Illumina MiSeq (Illumina, San Diego, CA, USA).

For short-read sequencing, a single colony was inoculated in 4 mL Mueller Hinton broth and incubated overnight at 35–37˚C. The MasterPure Complete DNA & RNA Purification kit (Epicentre, Madison, WI, USA) was used to extract genomic DNA. Libraries were prepared using the Nextera XT sample preparation kit (Illumina, San Diego, CA, USA) and sequenced with 2x 250 bp paired end sequencing using the Illumina MiSeq platform (Illumina, San Diego, CA, USA).

For long-read sequencing, high-molecular-weight DNA was isolated from fresh overnight cultures. Briefly, a single bacterial colony was inoculated in 10 mL Mueller-Hinton broth and incubated overnight at 35–37˚C. DNA was extracted using the MagAttract HMW DNA kit (Qiagen, Hilden, Germany) according to the manufacturer's protocol. DNA concentrations were measured using Qubit fluorometer (Thermo Fisher Scientific, Waltham, MA, USA). Sequencing libraries were prepared using the SMRTbell Express Template Prep kit 2.0 (Pacific BioSciences, CA, USA) and whole-genome sequencing was performed on the PacBio Sequel I using the Sequel Sequencing kit 3.0 (Pacific BioSciences, CA, USA). The sequences were submitted to NCBI under BioProject PRJNA927131. An overview of the sequenced isolates and their phenotypic and genotypic characteristics is provided in S3 Table.

## De novo assembly, genotyping and phylogenetic analysis

Short-read data was trimmed with TrimGalore v.0.4.4 (https://github.com/FelixKrueger/TrimGalore) and assembled *de novo* using SPAdes v.3.13.0 [25] built within BacPipe v1.2.6 [26]. Assembly of long-read sequencing data was done using HGAP with default parameters, included in SMRT Link v10.1 (Pacific BioSciences, CA, USA). Assembly quality was assessed with Quast [27]. The assembled genome was annotated using Prokka v.1.12 [28]. Additional analysis was performed using BacPipe v1.2.6 including the PubMLST database [29], ResFinder (database 2022-05-24) [30], virulence factor database (VFDB) [31] and PlasmidFinder (database 2021-11-29) [32] and PointFinder (database 2021-02-01) [33]. Species identification was confirmed based on WGS data using PubMLST [29]. Kleborate 2.2.0 was used to genotypically characterize *Klebsiella* spp. [34].

Colistin-susceptible strains used as a reference for detection of colistin resistant mutations are listed in S4 Table. For all isolates, mutations in the *pmrAB* and *phoPQ* two-component systems were determined. For *E. coli*, mutations in *pmrC*, *pmrD* and *mgrB* were additionally investigated. For *K. pneumoniae*, mutations in *pmrC*, *pmrD*, *mgrB* and its promotor region, *crrAB*, *yciM*, *lpxM* and *arnA* were additionally explored. For *Enterobacte*r spp., mutations in *mgrB* and its promotor were also considered [11,35]. Virulence genes were functionally classified according to the VFDB [31].

For core genome multilocus sequence typing (cgMLST), a gene-by-gene approach was utilized by developing a tailor-made scheme for the specific study, and subsequently assessing allelic loci distances using ChewBBACA [36]. Clonal relatedness was defined as ≤10, ≤11 or ≤12 allelic differences between isolates of *E. coli* [37,38], *Enterobacter* spp. [39] and *Klebsiella* spp. [39], respectively. Trees were visualized using Grapetree [40].

## Statistical tests and visualization

Statistical tests and visualization were performed using R version 4.2.0 [41]. Differences in proportions of colistin resistance between the first and second measurement per One Health sector and country were tested using generalized linear models with a negative binomial distribution. Clustering within wards or units was taken into account. Associations between colistin use and resistance in livestock farms were assessed using a generalized linear model. The association between the presence of an iron uptake system and animal-or human-derived isolates was tested with the Fisher's exact test. P-values of <0.05 were considered statistically significant.

## Ethics statement

The study protocol was review by the Medical Research and Ethics Committee of the University Medical Center Utrecht (Utrecht, the Netherlands) (Protocol Number 17-426/C) and the Ethics Committee of the University Hospital Antwerp (Antwerp, Belgium) (Belgian Registration Number B300210733784), the Medical Research and Ethics Committee of the Maastricht University Medical Center+ (Maastricht, the Netherlands) (METC 2017–0115 and METC 2017–0116), the Ethics Committee of the University Hospitals Leuven (Leuven, Belgium) (S61807 and S61353). The study was judged to be beyond the scope of the Medical Research Involving Human Subjects Act and the Belgian Law on Experiments on Humans, dated May 7[th], 2004. Written (in Belgium, via information and consent form) or verbal (in the Netherlands) informed consent for data collection and taking the fecal, perianal or gastrointestinal stoma swab for microbiological culture was obtained from all participants or their legal representatives. The authors did not have access to information that could identify individual participants during or after data collection.

Approval by an animal welfare body was not required. The procedure to collect fresh fecal droppings is considered to cause no discomfort, and animals were neither handled nor sacrificed during the study (EC Directive 2010/63). All data were anonymized, i.e. data cannot be directly or indirectly related to their source. Data on institutions and farms were pseudonymized, i.e. identifying information replaced by a code and a key file that links this code to the identifying information is kept separate from the research data.

## Results

### Presence of colistin-resistant Enterobacterales in hospitals, long-term care facilities, day care centres and farms in Belgium and the Netherlands

Of the 1268 Enterobacterales isolates picked from the selective colistin agar plate, 748 (58.9%) were confirmed as colistin resistant (MIC ≥4 mg/L). These colistin-resistant isolates were

distributed in 24 bacterial species, the majority being *Escherichia coli* (63.2%), followed by *Klebsiella* spp. (22.5%), three quarter of which were *Klebsiella pneumoniae*, and 10.0% of *Enterobacter* spp. A larger variety in bacterial species was carried by humans compared to livestock animals (S1 Fig).

ColR-E isolates were found in all investigated One Health sectors, albeit with different frequency of occurrence by sector (Fig 1A). Each measurement, the percentage of patients carrying ColR-E at one Belgian hospital (7/160 (4.4%) and 16/202 (7.9%)) was significantly lower compared to the prevalence observed among patients at two Dutch hospitals (43/382 (11.3%) and 30/254 (11.8%)) (p<0.001) (Table 1). Similar occurrences were observed between the two Dutch hospitals and the two measurements (9.1–12.2%).

On the other hand, the prevalence of ColR-E colonization was significantly higher in Belgian LTCF residents (67/656, 10.2%) as opposed to their Dutch counterparts (43/774, 5.6%). A total of 11/13 Belgian LTCF and 14/17 Dutch LTCF were positive for ColR-E with up to 21.6% and 16.7% of the residents colonized within a Belgian and Dutch LTCF, respectively. Similarly, the ColR-E colonization rate was higher in children attending day care centres in Belgium (79/448, 17.6%) than in those attending similar facilities in the Netherlands (64/499, 12.8%). Fifteen out of seventeen Belgian and 22/28 Dutch day care centres were ColR-E positive with up to 35.7% and 31.6% of the children colonized in a Belgian and Dutch day care centre, respectively.

The lowest occurrences were detected in the broiler farms in Belgium and the Netherlands. Each measurement, a larger proportion of the broilers were colonized in the Netherlands (20/380 (5.3%) and 6/390 (1.5%)) compared to Belgium (6/399 (1.5%) and 3/450 (0.7%)). ColR-E isolates were detected in 3/15 Belgian broiler farms. Within-farm occurrences ranged from 0 to 10% in the first and 0 to 3.3% in the second measurement. The number of Dutch broiler farms positive for ColR-E declined from 9/14 in the first measurement to 4/13 farms in the second measurement. Within-farm occurrences in the Dutch broiler farms ranged from 0 to 16.7% in the first measurement and from 0 to 10% in the second measurement.

The proportion of positive samples was higher in the Belgian pig farms than in the Dutch pig farms at both measurements: 87/399 (21.8%) and 98/420 (23.3%) vs 48/328 (14.6%) and 40/450 (8.9%), respectively. However, the percentage of positive samples varied greatly between different pig farms (0%-93.3% in Belgium and 0–46.7% in the Netherlands) (Fig 1B). Two Belgian pig farms showed consistently high occurrence of colistin resistance (≥70%) over a period of one year. On the other hand, ten Belgian broiler farms, one Dutch pig farm and four Dutch broiler farms showed no colistin resistance over the two measurements.

When investigating carriage of indicator bacteria (*E. coli*, *Klebsiella* spp. and *Enterobacter* spp) individually, few ColR-*E. coli* were detected in Belgian hospitalized patients (1.3%) compared to Dutch patients (7.1%) in the first measurement. The percentage of hospitalized patients carrying MDR isolates was similar in Belgium and the Netherlands (3.8–5.9%), while slightly higher percentages of older adults (3.2%) and children (6.3%) carried MDR isolates in Belgium compared to those in the Netherlands (1.9% of the older adults and 3.4% of the children). Similarly, MDR isolates were more prevalent in Belgian pigs (18.1–19.1%) compared to Dutch pigs (8.8–7.1%) in both measurements (S2 Fig).

## Colistin use in broiler and pig farms

In the study period, colistin treatment incidence was higher in the pig populations in comparison to broiler chickens. Among the surveyed farms, nearly all Belgian (14/15) and the majority of Dutch (11/15) pig farms employed colistin as a treatment six months before or during the study period (Fig 1B). In contrast, its use was limited to only one Belgian and two Dutch

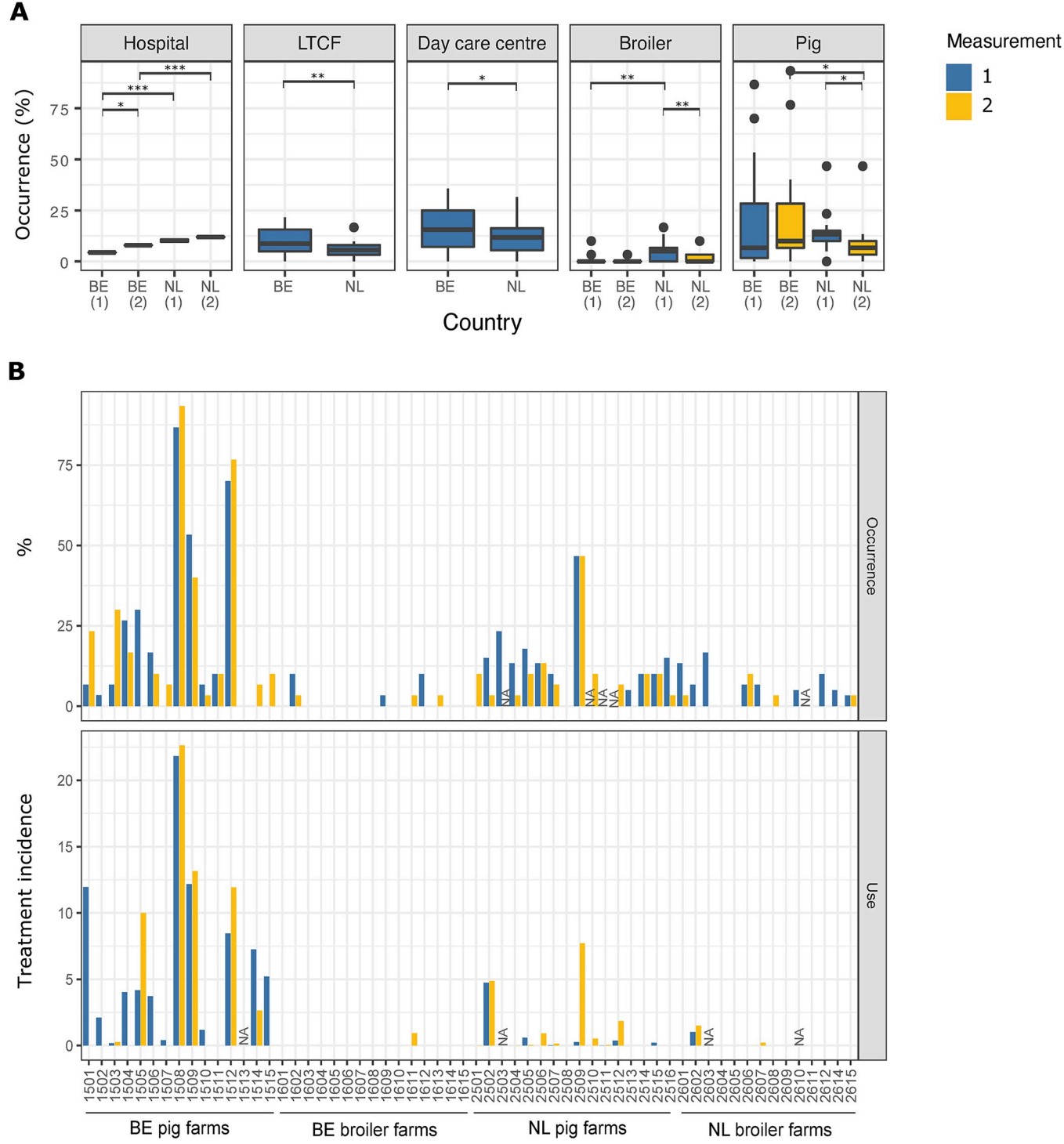

**Fig 1.** Occurrence of colistin resistance in One Health sectors (A) and colistin treatment incidence in farms (B). (A) Boxplots of the occurrence of colistin-resistant Enterobacterales in hospitalized patients, individuals in day care centres and long-term care facilities, broilers and pigs. Differences in the occurrences of colistin resistance were tested using generalized linear models with negative binomial distribution. * (p<0.05), ** (p<0.01), *** (p<0.001). (B) Occurrence of colistin-resistant Enterobacterales and colistin treatment incidence per farm. Colistin treatment incidence includes prescriptions one year before the first measurement (1) and between the first and second measurement (2). BE: Belgium, NL: The Netherlands, LTCF: Long-term care facility, NA: Data not available.

**Table 1. Comparison of colistin resistance between Belgium and the Netherlands by measurement and by sector.**

| Sector (measurement) | Number of samples | | Colistin resistance (%) (range of within site percentage of positive samples) | | Number of positive sites | | Risk difference (%) | 95% CI | p-value |
|---|---|---|---|---|---|---|---|---|---|
| | BE [a] | NL[b] | BE [a] | NL [b] | BE [a] | NL [b] | | | |
| Hospital (1) | 160 | 382 | 4.4 | 11.3 (9.1–11.3) | 1/1 | 2/2 | 6.9 | 3.9–9.9 | *** (<0.001) |
| Hospital (2) | 202 | 254 | 7.9 | 11.8 (11.7–12.2) | 1/1 | 2/2 | 3.9 | 2.2–5.5 | *** (<0.001) |
| LTCF | 656 | 774 | 10.2 (1.9–21.6) | 5.6 (0–16.7) | 11/13 | 14/17 | -4.7 | -7.9 - -1.4 | ** (<0.01) |
| Day care | 448 | 499 | 17.6 (0–35.7) | 12.8 (0–31.6) | 15/17 | 22/28 | -4.8 | -9.6 - -0.1 | * (<0.05) |
| Broiler (1) | 399 | 380 | 1.5 (0–10) | 5.3 (0–16.7) | 3/15 | 9/14 | 3.8 | 0.9–6.6 | ** (<0.01) |
| Broiler (2) | 450 | 390 | 0.7 (0–3.3) | 1.5 (0–10) | 3/15 | 4/13 | 0.9 | -0.6–2.4 | ns [c] |
| Pig (1) | 399 | 328 | 21.8 (0–86.7) | 14.6 (0–46.7) | 11/15 | 11/13 | -7.2 | -18.0–3.7 | ns [c] |
| Pig (2) | 420 | 450 | 23.3 (0–93.3) | 8.9 (0–46.7) | 12/14 | 12/15 | -14.4 | -26.8 - -2.1 | * (<0.05) |

Differences in the proportions of colistin resistance between Belgium and the Netherlands were tested using generalized linear models with a negative binomial distribution. P-values of <0.05 were considered statistically significant. [a]BE: Belgium, [b]NL: Netherlands, [c]ns: Not significant.

*** (p<0.001)

** (p<0.01)

* (p<0.05).

broiler farms. Notably, the colistin treatment frequency across the livestock farms displayed high variability on a per-farm basis. In particular, three Belgian pig farms (farm IDs 1508, 1509 and 1512) showed high colistin use during and between the measurement periods which was linked to a high occurrence of colistin resistance (>50% of the pigs positive for carriage of colistin-resistant Enterobacterales) (Fig 1B). Colistin resistance was positively associated with the prior use of colistin within pig farms (Table 2).

**Table 2. Association of colistin resistance with prior colistin use in pig farms in Belgium (n = 14) and the Netherlands (n = 15).**

| Measurement | Colistin use | Country | Estimated change in odds of colistin resistance for each unit increase in colistin use[a] | 95% CI | p-value |
|---|---|---|---|---|---|
| Measurement 1 | 1 year before measurement | Belgium | 1.13 | 1.04–1.23 | * (<0.05) |
| | | Netherlands | 1.08 | 0.71–1.66 | ns[b] |
| Measurement 2 | 2–3 years before measurement | Belgium | 1.12 | 1.02–1.22 | * (<0.05) |
| | | Netherlands | 0.87 | 0.47–1.61 | ns[b] |
| | 6 to 15 months before measurement | Belgium | 1.22 | 1.09–1.36 | ** (<0.01) |
| | | Netherlands | 1.85 | 1.20–2.86 | * (<0.05) |
| | 6 months before measurement | Belgium | 1.18 | 1.03–1.35 | * (<0.05) |
| | | Netherlands | 1.39 | 1.09–1.76 | * (<0.05) |

A total of 379 and 420 Belgian as well as 298 and 450 Dutch pigs were screened for the carriage of colistin-resistant Enterobacterales in the first and second measurements, respectively. Associations were assessed using a generalized linear model.

[a] The estimated change in odds represents the odds of colistin resistance after colistin use compared to the odds of colistin resistance without colistin use.

[b] ns: Not significant.

## Chromosomal and plasmid-mediated colistin resistance detected in colistin-resistant *E. coli*, *Klebsiella* spp. and *Enterobacter* spp.

A total of 343 *Escherichia coli*, 112 *Klebsiella pneumoniae*, 28 *Enterobacter (quasi)roggenkampii*, 24 *Klebsiella variicola*, 13 *Enterobacter cloacae*, 10 *Enterobacter asburiae*, 8 *Enterobacter kobei*, 6 *Klebsiella michiganensis*, 5 *Enterobacter hormaechei*, 5 *Enterobacter ludwigii*, 4 *Klebsiella quasipneumoniae*, 2 *Klebsiella aerogenes*, 2 *Klebsiella oxytoca* were sequenced to study the molecular make-up of colistin-resistant isolates.

Overall, mutations were most prevalent in *pmrB* (440/562, 78.3%), followed by *pmrA* (222/562, 39.5%) and *phoQ* (186/562, 33.1%). Mutations in *phoP* were less prevalent (27/562, 4.8%) (S3 Table). Alterations in *mgrB* or its promotor region were detected in *E. coli* (76/343, 22.2%), *Enterobacter* spp. (32/69, 46.4%) and *Klebsiella* spp. (93/150, 62.0%) (S5 Table). Concurrent mutations in two component system PmrAB and PhoPQ or its regulators were present in most isolates (508/571), however, single mutations led to colistin resistance in 49 isolates (8.7%) (S6 Table).

Plasmid-mediated *mcr*-genes were detected in 36 of the 562 sequenced colistin-resistant isolates (6.4%). The *mcr* genes were detected in 31/178 (17.4%) of the screened pigs, none of the broilers, 1/96 (1.0%) of the hospitalized patients, 2/112 (1.8%) of the residents in LTCF and 1/146 (0.7%) of the children. Bacterial species were 31 *E. coli* (83.8%), 1 *E. asburiae* (2.7%), 1 *E. roggenkampii* (2.7%), 2 *E. kobei* (5.4%) and 1 *E. hormaechei* (2.7%). Plasmid-mediated colistin resistance genes were not detected in any *Klebsiella* species isolates. Genes *mcr-1.1*, *mcr-2.1*, *mcr-2.2* and *mcr-5.1* were all detected in *E. coli* isolated from Belgian pig farms, while *mcr-9* and *mcr-10* were detected in *Enterobacter* isolates from hospitalized patients and healthy individuals (from a Belgian hospital, day care center and LTCF, and a Dutch hospital).

Different MGEs were flanking these *mcr*-genes: IS26 flanked *mcr-1.1*, *mcr-5.1*, *mcr-9.1* and *mcr-10*, ISApl1 flanked *mcr-1.1*, while *mcr-2* was flanked by IS*Ec*69. The presence of *mcr-1.1* and *mcr-2.1* genes was observed on IncX4 and IncHI2 plasmids, while the *mcr-5.1* genes could be identified on an IncFII (29) plasmid (Fig 2). Aligning the reads to the most similar reference plasmid sequence according to blastn, showed that several *mcr-1.1*-harboring sequences from Belgian pig farm 12 were highly similar (query coverage 100%, >99.70% identity) to pMFDS2258.1 (accession number MK869757.1), a plasmid isolated from chicken meat from Brazil in 2017 (S3A Fig). Similarly, an *mcr-1*-haboring plasmid from pig farm 7 was aligned to a plasmid from an Italian stream (accession number MF449287.1) (S3D Fig). Other *mcr-1.1* plasmid sequences from Belgian pigs could be aligned to various IncX4 and IncHI2 plasmids with lower query coverages (1%-77%) (S3B–S3E Fig). In addition, high query coverage (>99%) was found between *mcr-2*-harboring sequences from Belgian pig farms 4 and 9 to pKP37 (accession number LT598652.1), an *mcr-2.1*- carrying plasmid isolated from Belgian pigs in 2016 [42] (S3F Fig). Both the *mcr-1*- and the *mcr-2*-harboring plasmids were detected over time in the same pig farms, suggesting the persistence of these plasmids in the farms over a period of one year. *mcr-5.1* sequences from Belgian pigs were aligned to a plasmid from human stool in Mexico (pYU07-18_89; CP035549.1, query coverage 95%) and from pork meat in Vietnam (pVE155; AP018354.1, query coverage 57%) (S3G and S3H Fig). *mcr-9* and *mcr-10-* containing plasmids showed similarities with plasmids previously isolated in Egypt, Spain and China (query coverage 0.7–89%) (S3I–S3L Fig).

## Phenotypic and genotypic resistance identified in colistin-resistant isolates in various One Health sectors

Phenotypic MDR was detected in 61.5% (291/473) of *E. coli*, 33.1% (55/166) *Klebsiella* spp. and 78.7% (55/75) of *Enterobacter* spp. isolates. Colistin-resistant and MDR human isolates

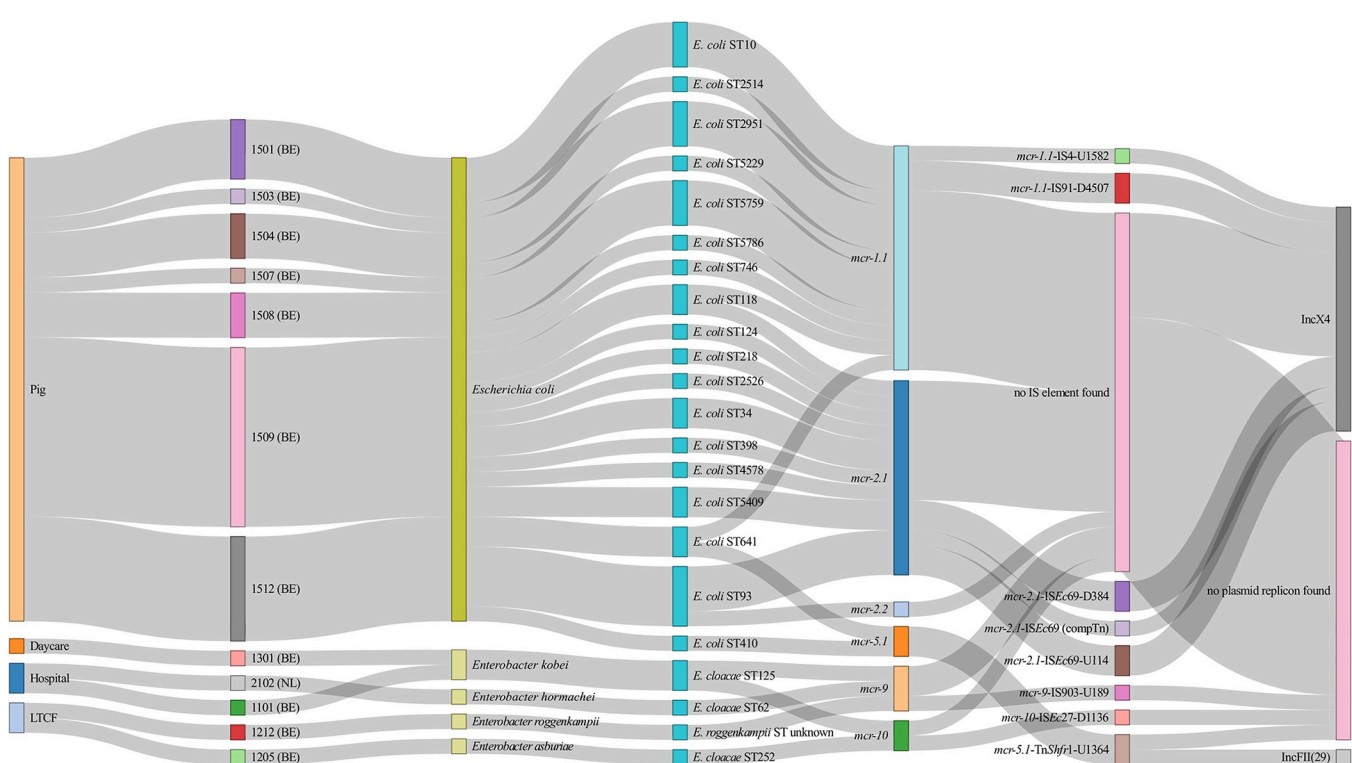

**Fig 2. Sankey diagram of the origin and genetic context of *mcr*-genes.** The closest IS element to the *mcr* gene is indicated together with the upstream (U) or downstream (D) and distance to the *mcr* gene. The width of the lines in the diagram is proportional to the number of isolates. LTCF: Long-term care facility, BE: Belgium, NL: The Netherlands, compTn: Composite transposon.

were most commonly resistant to ampicillin (46.8% of the human MDR isolates), to amoxicillin- clavulanic acid (71.2%) and to cefoxitin (43.2%). MDR livestock-derived isolates were regularly resistant to ampicillin (80.0% of broiler MDR isolates, 83.0% of the porcine MDR isolates) and to trimethoprim-sulfamethoxazole (76.0% of broiler MDR isolates, 83.0% of the porcine MDR isolates). The percentage of livestock-derived *E. coli* (82.4%) and *Klebsiella* spp. (58.5%) isolates with an MDR phenotype was higher compared to human isolates (30.2% of *E. coli* and 16.8% of *Klebsiella* spp.). For *Enterobacter* isolates, this difference in MDR proportions was not observed (83.3% of the animal and 78.3% of the human isolates) (Fig 3A). Nonetheless, the majority of the colistin-resistant *E. coli* (419/473, 88.6%), *Klebsiella* spp., (126/166, 75.9%) and *Enterobacter* spp. (50/75, 66.7%) were phenotypically susceptible to the critically important antibiotics (fluoroquinolones, extended-spectrum cephalosporins, carbapenems and aminoglycosides). Carbapenem resistance and carbapenemase genes were not found in any isolate of the different settings. Phenotypic resistance rates to extended-spectrum cephalosporins, fluoroquinolones or aminoglycosides were relatively low (7.2%, 6.2% and 2.9% of the isolates, respectively) (Fig 3B). Acquired ESBL genes were detected in 2.8% of the isolates, *qnr* genes were detected in 6.9% isolates and mutations in the quinolone-resistance determining regions (QRDR) were detected in 10.3% of the isolates (Fig 4).

### Associated resistance in colistin-resistant *E. coli*

Resistance to extended-spectrum cephalosporins was detected in 4.0% (19/473) of the *E. coli* isolates from all sectors, except from Dutch pig farms. ESBL genes (*bla*CTX-M) were acquired by 2.6% (9/343) of the sequenced *E. coli*. Phenotypic resistance to ciprofloxacin was detected

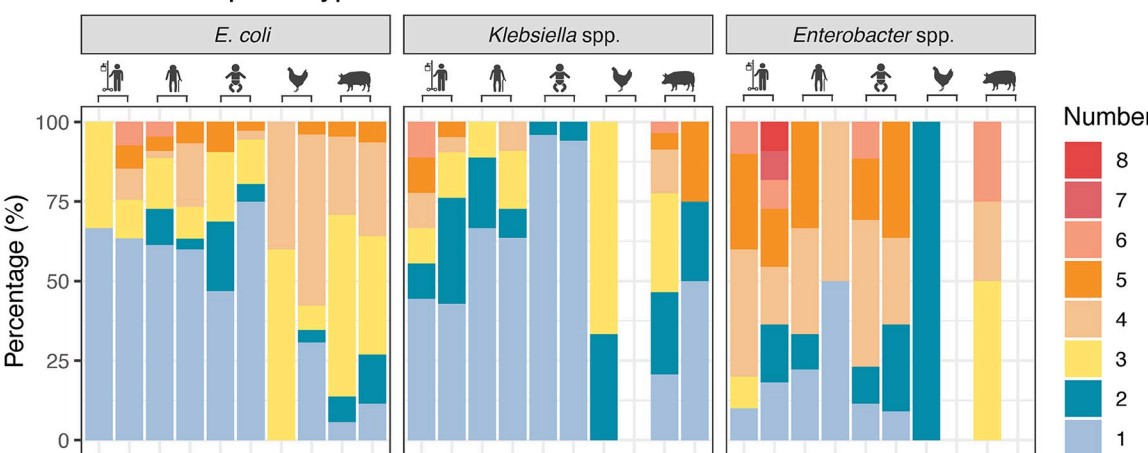

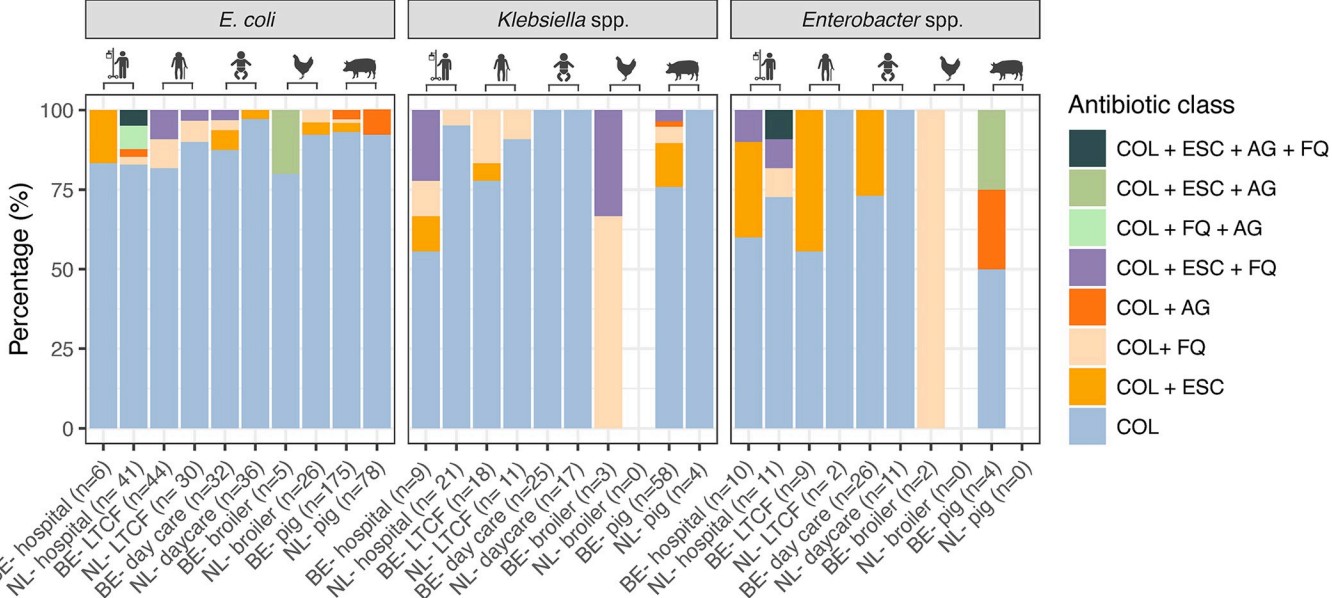

**Fig 3. Phenotypic antibiotic resistance of colistin-resistant Enterobacterales.** (A) Stacked barplots of the proportion of isolates phenotypically resistant to a number of antibiotic classes. (B) Stacked barplots of the proportion of isolates phenotypically resistant to critically important antibiotics. AG: Aminoglycosides, COL: Colistin, ESC: Extended-spectrum cephalosporins, FQ: Fluoroquinolones, LTCF: Long-term care facility, BE: Belgium, NL: The Netherlands.

in 6.8% (32/473) of the isolated *E. coli*. Plasmid-mediated *qnr* genes were detected in two hospitalized patients from the Netherlands and two children, one LTCF resident and twelve pigs from Belgium. A total of 12.2% (42/343) harbored one or more mutations in *gyrA*, *parC* and/or *parE* (Fig 4). Phenotypic aminoglycoside (gentamicin, tobramycin or amikacin) resistance was present in *E. coli* from Dutch hospital patients, Belgian broilers and Belgian and Dutch pigs (3.6%, 17/343). The *aac(3)* family resistance genes was present in 4.1% (14/343). Combined resistance to colistin, extended-spectrum cephalosporins, fluoroquinolones and aminoglycosides was detected in 2 *E. coli* isolates (0.4%) from Dutch hospitals.

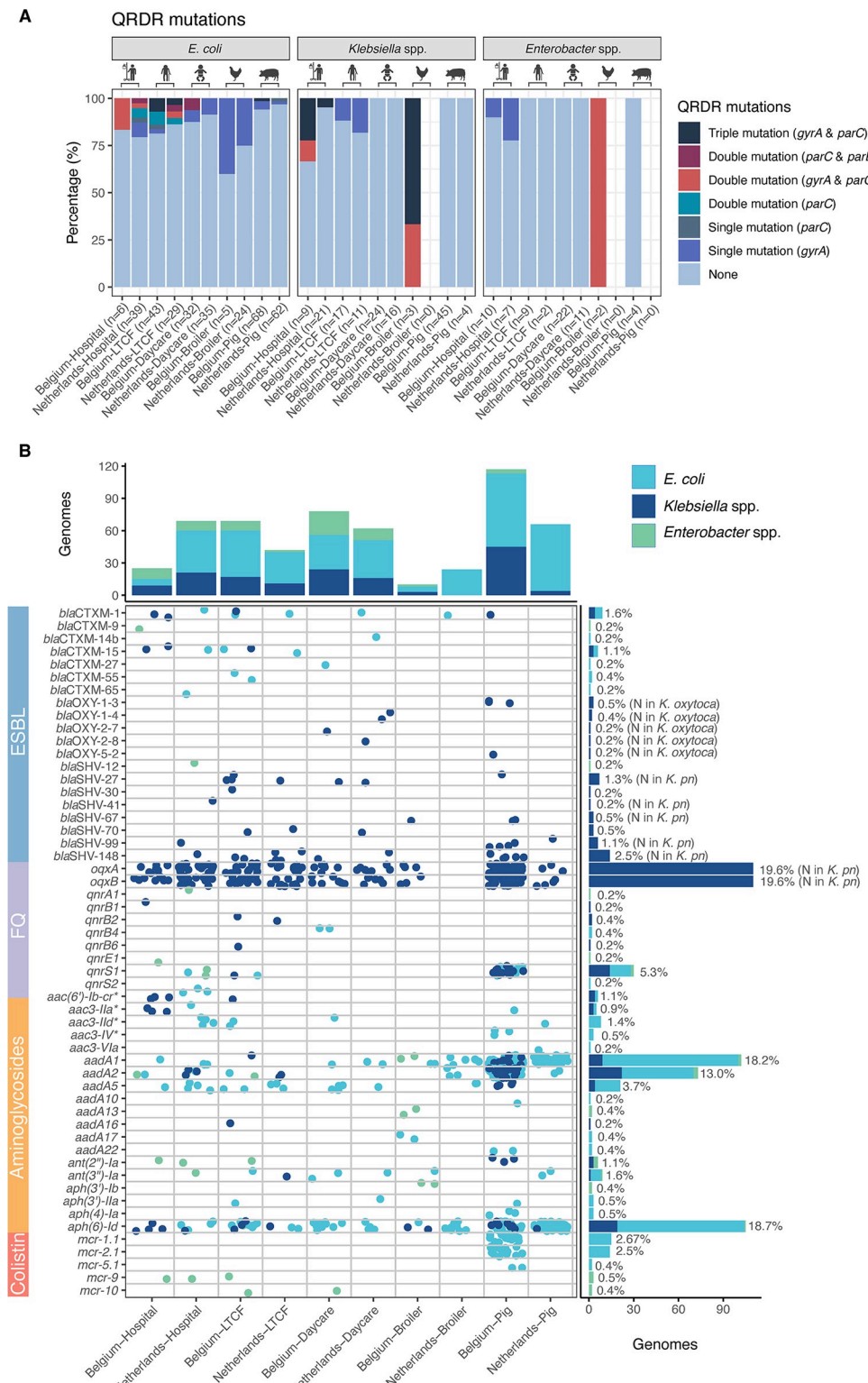

**Fig 4.** Genotypic fluoroquinolone resistance mutations (A) and resistance genes (B) for critically important antibiotics detected in colistin-resistant Enterobacterales. (A) Stacked barplots of the proportion of colistin-resistant *E. coli* and *Klebsiella* spp. with mutations in the quinolone-resistance determining regions (QRDR) linked to fluoroquinolone resistance. (B) Resistance genes for critically important antibiotics detected in colistin-resistant Enterobacterales. Each circle represents a genome (isolate) colored by species. Barplots show the number of genomes from the different

sectors (top) and containing the resistance gene (right) colored by species. N: Naturally occurring genes, *Kpn*: *K. pneumoniae*, QRDR: Quinolone-resistance determining region, FQ: Fluoroquinolone, ESBL: Extended-spectrum beta-lactamase, * aminoglycoside genes linked to resistance to gentamicin, tobramycin.

## Associated resistance in colistin-resistant *Klebsiella* spp

Acquired ESBL genes were detected in 3.3% (5/150) of the *Klebsiella* isolates. Ciprofloxacin resistance was present in 16/166 isolates (9.6%). This resistance was linked to mutations in QRDR regions of *gyrA* and *parC* in 11/150 isolates (7.3%) and *qnr* genes in 18/150 isolates (12.0%). A total of 48/150 isolates (32.0%) harbored aminoglycoside resistance genes (Fig 4).

## Associated resistance in colistin-resistant *Enterobacter* spp

An intermediate phenotype for meropenem was observed in *E. cloacae* from one Belgian broiler and to imipenem from one Dutch child (0.2%). Resistance to extended-spectrum cephalosporins was detected in 24.0% (18/75) *Enterobacter* isolates, while the proportion of isolates resistant to ciprofloxacin (8.0%, 6/75) and aminoglycosides (4.0%, 3/75) was low (Fig 3). A single mutation in QRDR region of *gyrA* (S83I or S83Y) was detected in 3 isolates (7.2%) from hospitalized patients and both Belgian broiler isolates harbored a mutation in *gyrA* (S83I) and *parC* (S80I). ESBL genes among *Enterobacter* spp. were uncommon: $bla_{CTX-M-9}$ was harbored by a *E. kobei* isolate from a Belgian patient and $bla_{SHV-12}$ was harbored by a *E. hormaechei* isolate from a Dutch patient (Fig 4B).

## Virulence potential of colistin-resistant *E. coli*, *Klebsiella* and *Enterobacter* isolates from different One Health sectors

Virulence factors present in all isolates were linked to fimbrial adhesins, inflammatory signaling, invasion and the enterobactin siderophore. Various iron uptake systems such as aerobactin, salmochelin and yersiniabactin were associated mainly with human *Escherichia* isolates and were less prevalent among livestock isolates (p<0.001) (S4 Fig).

Colistin resistance was also detected in *K. pneumoniae* harboring hypervirulence genes and various *E. coli* pathotypes, suggesting that these commensal bacteria may have pathogenic potential. Investigation of virulence-associated genes have uncovered the presence of virulence plasmid-associated loci, specifically *iuc*, *iro*, and *rmpA/rmpA2*, in three colistin-resistant *K. pneumoniae* strains. These strains have the potential to exhibit hypervirulent characteristics and belong to two distinct sequence types: ST5 (K39, O1 type), originating from two separate swine farms in Belgium, and ST592 (K57, O3b type) obtained from a medical facility in the Netherlands.

Colistin resistance was detected in different pathotypes including intestinal and extraintestinal pathogenic *E. coli*. Colistin-resistant *E. coli* pathotypes detected were shiga-toxin producing *E. coli* (STEC, porcine *E. coli* n = 2), diffusely adherent *E. coli* (DAEC, human-derived *E. coli*, n = 14), atypical enteropathogenic *E. coli* (aEPEC, n = 18 from all One Health sectors) and uropathogenic *E. coli* (UPEC) harboring *papC*, *papG* and *iucC* (human-derived *E. coli*, n = 24). Of these pathogenic *E. coli*, 22 isolates (35.5%) were MDR. Half of these pathogenic, MDR *E. coli* (n = 11) belonged to known invasive extraintestinal *E. coli* STs (ST10, ST38, ST69, ST73 and ST131) [43].

## Detection of colistin resistance within pandemic lineages

A diversity of STs was detected among the ColR-E isolates. Several high risk clonal lineages, such as *E. coli* ST10 (n = 35) of which three harbored *mcr-1.1* on an IncX4 plasmid, ST38

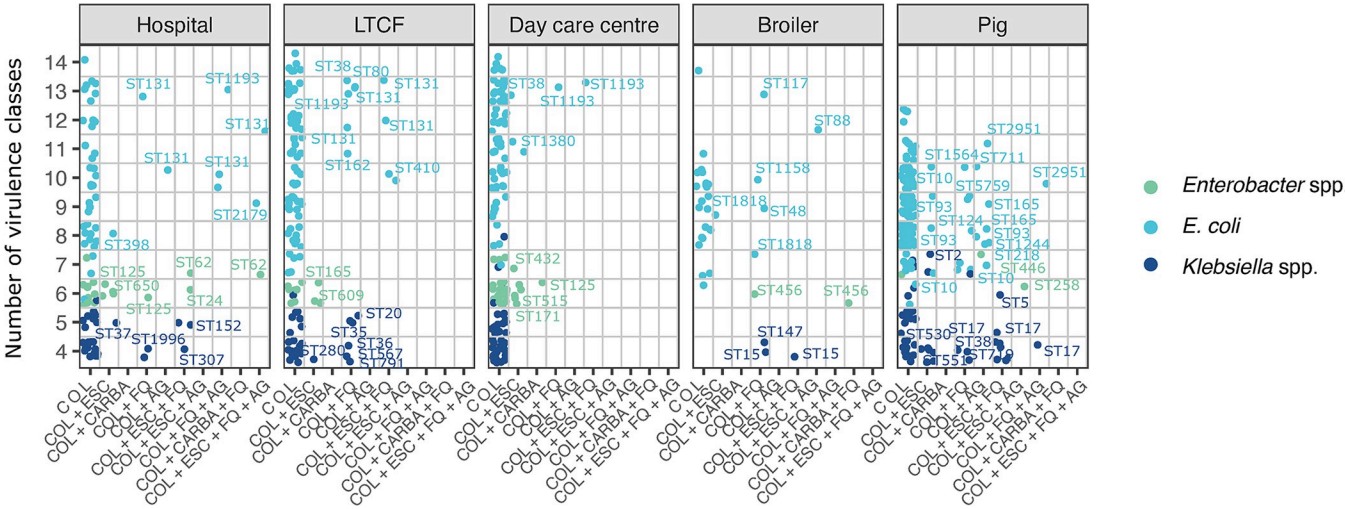

**Fig 5. Insights into the resistance to critically important antibiotics and the number of virulence classes present in colistin-resistant isolates from different One Health sectors.** (A) Dotplot of the resistance and number of virulence classes per isolate. Each circle represents a single isolate. Color indicates species and sequence types of resistant isolates are indicates with labels. LTCF: Long term care facility.

(n = 7), ST131 (n = 17), ST405 (n = 2), ST648 (n = 2), ST1193 (n = 4), *K. pneumoniae* ST15 (n = 2), ST45 (n = 7), ST101 (n = 1), ST147 (n = 1) and ST307 (n = 1), and *E. cloacae* ST171 (n = 1) were detected. Most of these isolates (n = 57/80, 71.3%) were not resistant to fluoroquinolones, extended-spectrum cephalosporins, carbapenems and aminoglycosides. However, human-derived *E. coli* ST131 and ST1193 showed high virulence potential combined with resistance to critically important antibiotics ([Fig 5]). Within LTCFs and hospitals, the presence of *E. coli* ST131 strains displaying colistin resistance and possessing $bla_{CTX-M-15}$ and fluoroquinolone resistance mutations (H30Rx) were discovered (n = 3), along with the detection of ST131-H30R (fluoroquinolone-resistant) strains (n = 3).

## Potential transmission pathways of ColR-E across and within the One Health framework

Inter-host transmission between humans and livestock animals (animal-to-human transmission) was not detected. However, clusters of related isolates were detected within sampling sites (animal-to-animal and human-to-human transmission), indicating that transmission of ColR-E occurred within broiler farms and within pig farms, between children within the day care centres, and between patients residing in the LTCFs and the hospitals ([Fig 6] and [S7 Table]). Related isolates were also detected between different sampling sites. Closely related isolates of *K. pneumoniae* ST45 (n = 5) were detected between the Dutch and Belgian hospitals, a Belgian day care centre and a Dutch LTCF. Similarly, a clonal clustering of *E. coli* ST10 (n = 4) was identified at two Belgian day care centres and at a Dutch day care centre. The transmission of *mcr-1.1-* and *mcr-2.1-* harboring *E. coli* was also detected amongst Belgian pig farms. A recurrent presence of clonally-related strains was noted during both measurements, strongly suggesting the persistent circulation of these particular isolates within the pig farm ecosystem ([S7 Table]).

## Discussion

Using an integrative approach, this study showed the presence of ColR-E among all studied One Health sectors and provides a detailed overview of the phenotypic and molecular makeup of these colistin-resistant isolates from different niches.

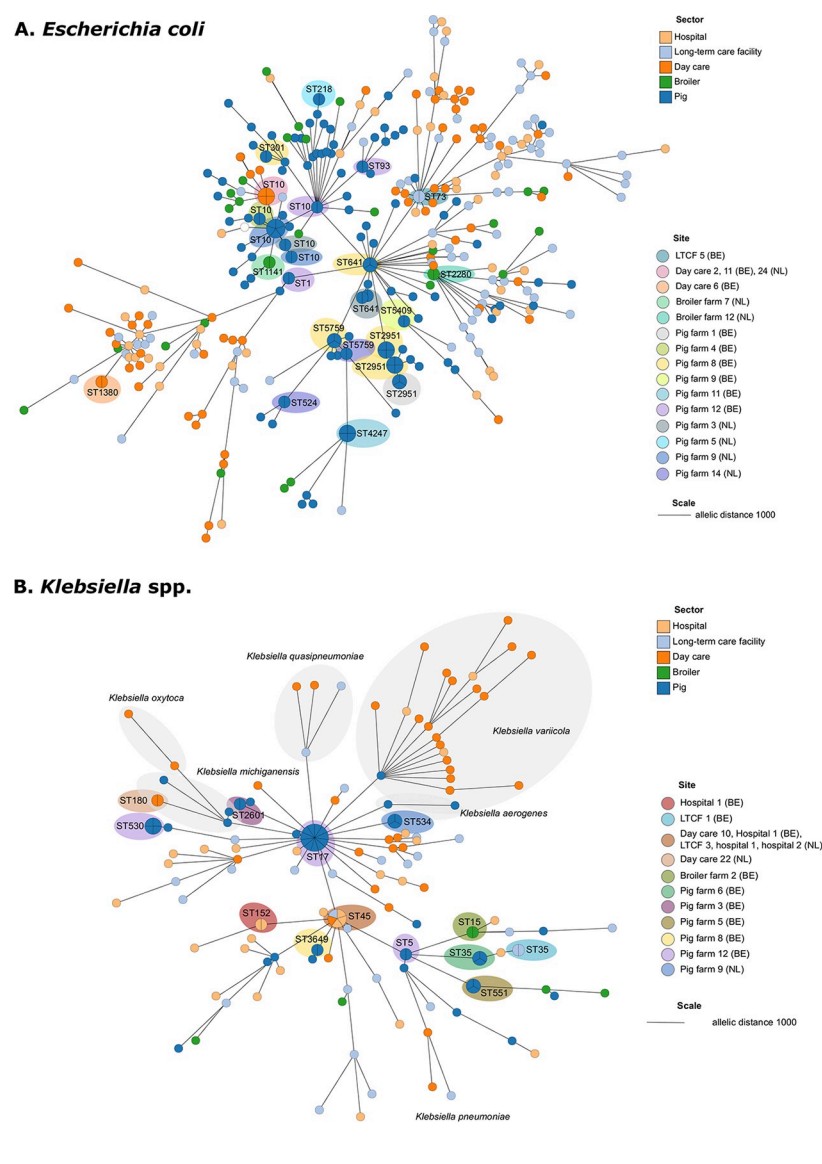

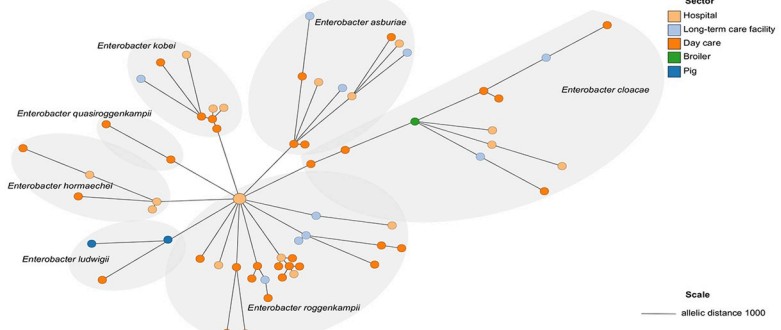

**Fig 6.** Minimum spanning trees of *Escherichia coli* (A), *Klebsiella* spp. (B) and *Enterobacter* spp. (C) isolated from humans in hospitals, long-term care facilities (LTCF), day care centres, broilers and pigs in farms. Minimum spanning trees based on allelic distances of cgMLST profile data (2967 loci for *E. coli*, 3362 loci for *Klebsiella* spp., and 2952 loci for *Enterobacter* spp.). Branch lengths indicate the allelic distance as indicated by the tree scale. Collapsed nodes indicate genetically related isolates with ≤10 and 12 allelic differences for *E. coli* and *Klebsiella* spp. respectively. The

sequence type is indicated for each cluster of related isolates. The origin of isolation is shown as colored nodes for each isolates.

This is the first study to have investigated colistin resistance in humans and animals in Belgium and the Netherlands using a One Health approach with a uniform methodology. In the Netherlands, the parallel monitoring of antimicrobial resistance and antibiotic use in animals and humans is reported within Nethmap-MARAN. However, colistin screening in humans is not included [17]. In Belgium, the BELMAP report summarizes the antibiotic use and resistance data in the human and veterinary sectors to provide a One Health overview of the Belgian situation [44]. These national reports lack whole genome sequencing of colistin-resistant isolates. The available studies on colistin resistance using a One Health concept essentially consist of systematic reviews and meta-analyses of available literature involving a limited number or specific settings (not using a One Health approach) and using different methodologies [13,45,46].

In this study, we estimated the prevalence of colistin resistance in various Belgian and Dutch One Health sectors, with the exception of the environment. Depending on the farm, the percentage of pigs within a farm colonized with ColR-E varied from 0% to 93.3%. The level of colistin resistance was positively associated with the prior colistin usage in these pig farms, as was also shown by other studies in food animals [47,48]. Although the sales of polymyxins in veterinary medicine is decreasing since 2011 [17,44,47], colistin was used in most of the pig farms in this study. Pigs remain the species with the largest use of colistin, especially weaner pigs for the treatment of enteropathogenic *E. coli* infections [44,49,50]. In contrast, colistin was used less frequently in the studied broiler farms which is reflected by the low percentage of broilers carrying ColR-E (2.2%). According to the national and European monitoring systems investigating resistance in indicator bacteria from healthy food-producing animals, prevalence of colistin resistance remained stable and very low (below 10%) over the years [44,49,51]. Colistin resistance in *E. coli* was not detected in the gastrointestinal tract of food-producing animals, meat and vegetables in the Netherlands in 2021 when using passive screening (non-selective isolation) [17]. The discrepancies with the prevalence found in pig farms in this study can probably be explained by the enrichment step and selective culturing methods we used here and which may have resulted in higher prevalence comparatively to studies using less sensitive methods [52,53]. Secondly, the selected farms had higher than average antibiotic use and are not representative for all farms in Belgium and in the Netherlands. Notwithstanding that the use of colistin in food-producing animals outweighs the use of colistin in humans in Europe [47], the prevalence of fecal carriage of ColR-E was relatively frequent in the three different human health sectors assessed in this study. The proportion of hospitalized patients carrying ColR-E was higher in the Netherlands (11.3–11.8%) than in Belgium (4.4–7.9%) though this can not be considered as representative for the whole country as only one Belgian and two Dutch hospitals were involved in the present study. In contrast, proportions of humans colonized with ColR-E in the other human health sectors were higher in Belgium (10.2% in LTCF and 17.6% in day care centres) compared to the Netherlands (5.6% in LTCF and 12.8% in day care centres). The occurrence of colistin resistance in the human population is sparsely studied in Europe. The prevalence of colistin resistance among human clinical Enterobacterales showed a regional variation of 2.4% to 3.4% in Europe [11]. In Switzerland, 1.5% of healthy individuals and 3.8% of primary care patients were carriers of ColR-E [54]. A recent study showed that 0.3% of the tested *E. coli* and 0.6% of the tested *K. pneumoniae* from clinical samples in the Netherlands were colistin-resistant [8] and colistin resistance in invasive clinical *E. coli* isolates from hospitalized patients in Belgium remains below 1% [44]. A surprisingly high

percentage of children in day care centres (15.1%) showed rectal carriage of ColR-E in this study. The causes of these high occurrences remain unclear. They could possibly partly be explained by factors investigated within our project, such as frequent contact of the studied children with animals (>70% of the children had contact with domestic animals, petting zoo animals and livestock animals) or hospital stays (7% of the Dutch children and 18% of the Belgian children were admitted in a hospital in the last six months before measurement) [55,56]. A total of 7% of the Belgian and 4% of the Dutch LTFC residents received antibiotic treatment in the last six months before measurement [57,58]. In addition, infection prevention measures (hand hygiene and a clean environment in LTCF as well as hand hygiene, cleaning of toys and avoiding fecal contamination such as cleaning the changing table, use of paper towels in day care centres) could be improved in most day care centres and LTCFs within the study to prevent the spread of resistant bacteria [55,57]. Adenosine triphosphate (ATP) measurements also showed higher levels of environmental contamination in Dutch hospitals compared to Belgian hospitals, likely due to differences in cleaning protocols [59].

Plasmid-mediated colistin resistance was detected in 6.4% of the isolates, which is in line with a previous study (9.7%) [8]. *mcr* genes were detected in 1.0% of the hospitalized patients, 1.8% of the LTCF residents, 0.7% of the children attending childcare centers, and none of the broilers which is lower than the estimated prevalences in these sectors worldwide (7.4% in healthy humans, 4.2% patients, 15.8% of chickens). The estimated prevalence in pigs was somewhat higher in our study (17.4%) compared to the meta-analysis (14.9%) [13]. Colistin resistance genes *mcr-1.1*, *mcr-2.1* and *mcr-5.1* were reported before in *E. coli* from Belgian pigs between 2012 and 2016 [60]. The persistence of the highly related IncX4 plasmids harboring *mcr-1.1* or *mcr-2.1* over a one-year period in these pig farms emphasizes the need for increased efforts to control the spread of *mcr* genes. For example, the ban on free use of colistin in animals has reduced the incidence of *mcr-1*-harboring IncX4-type plasmids, whose presence is associated with an effective dispersal potential in Enterobacterales and across different One Health niches (human, dogs, chickens and flies) [13–15,61]. Co-localization of *mcr* genes with other resistance genes on the same plasmid was not detected in our study, however, the majority of isolates was sequenced using short-read sequencing which complicates this analysis.

In this study, *mcr-1 to mcr-8* were not detected in human isolates, which is different from previous studies [13]. Reports on *mcr-9* in *K. pneumoniae* and *E. cloacae* from clinical samples in the Netherlands were published before (2015–2020) [8,62]. In this study, *mcr-9* and *mcr-10* were observed among several *Enterobacter* spp. human isolates from hospitals, day care centres and LTCF in Belgium and the Netherlands suggesting that surveillance of these *mcr* genes is needed. The *mcr*-harboring plasmids showed high levels of similarity to plasmids previously isolated in different countries worldwide showing the global spread of these *mcr*-harboring plasmids. In addition, *mcr* genes were flanked by IS elements, strongly suggesting the potential for mobility of these *mcr* genes.

In contrast to *mcr*-plasmids, chromosomal mutations in the core genome are found to be highly stable and irreversible, even after usage of colistin was stopped [4,11]. For the majority of the studied isolates, colistin resistance was caused by chromosomal mutations in genes/operons involved in the biosynthesis of the cell-wall LPS. The presence of these stable chromosomal mutations are worrying when present in key human pathogenic lineages. Indeed, various international high-risk clones, such as *E. coli* ST1193 and ST131 harbored chromosomal mutations, meaning that spread of colistin resistance is possible if these mutations are stable and transmitted to the descendants within that clone. In addition, genetically related clones of *K. pneumoniae* ST45 and *E. coli* ST10 were found at different sites, suggesting that these clones might have the potential to spread colistin resistance through the human population or were acquired by exposure to a common (food) source. Clusters of *E. coli* ST10 were also prevalent

in several pig farms some of which harbored the *mcr-1.1* (n = 3). *E. coli* ST10 was described as a reservoir for *mcr-1* genes before [63] and has the potential to disseminate this gene among food-producing animals.

Fortunately, inter-host transmission between humans and livestock animals was not observed in this study nor in other studies [61,64,65] and resistance to fluoroquinolones, extended-spectrum cephalosporins, aminoglycosides and carbapenems remained low (<10%), providing several alternative treatment options for these colistin-resistant isolates.

Our study has several limitations. Firstly, this study lacks extensive epidemiological data leaving gaps in our understanding of pathogen transmission. As a result strict thresholds for clonal relatedness were applied. Secondly, the chromosomal mutations were found by *in silico* analysis and were not experimentally confirmed. Thirdly, very few hospitals were included in the study and farms were not representative for the country as we selected farms with higher than average total antibiotic use making the occurrences of colistin resistance in these sectors not representative for the country. Nonetheless, to the best of our knowledge, this is the first One Health study to combine harmonized data on colistin use as well as phenotypic and molecular methods and provide detailed insights into the epidemiology of colistin resistance in the clinical setting, the community and livestock animals in Europe.

## Conclusion

Colistin resistance poses a significant threat to the treatment of MDR Gram-negative bacterial infections and its spread must be contained. The present research offers valuable insights into the presence of colistin resistance across various One Health sectors, which could inform containment strategies related to food production and prudent antibiotic use, with the aim of safeguarding public health.

## Supporting information

**S1 Fig. Distribution of the number of colistin-resistant Enterobacterales species by One Health sector and country.**
(PDF)

**S2 Fig.** Carriage of colistin-resistant Escherichia coli (A), Klebsiella spp. (B), Enterobacter spp. (C) and multi-drug resistant isolates (D) by humans and animals. The numbers indicated with the boxplots represent the total percentage of positive samples by country, measurement and sector. BE: Belgium, NL: the Netherlands, LTCF: long-term care facility.
(PDF)

**S3 Fig. Global view of BLAST comparisons between the *mcr*-harboring sequences and the most similar reference plasmid sequence (in grey) according to blastn.** Figure was generated using the BLAST Ring Image Generator (BRIG). Percentages indicate the query coverage of the *mcr*-containing sequence with the reference plasmid. Isolate ID, origin, insertion sequences, plasmid Inc type, resistance and virulence genes are indicated in different colors.
(PDF)

**S4 Fig. Virulence potential of ColR-E from different One Health sectors in Belgium and the Netherlands.** Heatmap of the percentage of ColR-E harboring virulence genes related to virulence classes (y-axis) per One Health sector in Belgium and the Netherlands (x-axis). Barplots show the number of genomes colored by species per virulence class (right) and colored by the number of virulence class per One Health sector (top). LTCF: long-term care facility.
(PDF)

**S1 Table. Overview of sampling dates, sites sampled per measurement, number of samples collected per measurement and presence of colistin-resistant Enterobacterales.**
(XLSX)

**S2 Table. Colistin use per farm expressed as TiDDDVet.**
(XLSX)

**S3 Table. Sequenced isolates and their phenotypic and genotypic characteristics.**
(XLSX)

**S4 Table. Genomes used as reference for the detection of mutations linked to colistin resistance.**
(XLSX)

**S5 Table. Alterations in *mgrB* or its promoter region in *Escherichia coli*, *Enterobacter* spp. and *Klebsiella* spp.**
(XLSX)

**S6 Table. Single alteration leading to colistin resistance (no other mutations in PmrAB and PhoPQ detected).**
(XLSX)

**S7 Table. Origin and characteristics of clonally related isolates.**
(XLSX)

## Acknowledgments

We are grateful to the farmers, the veterinarians and all collaborators of the participating farms for their contribution to the collection of the epidemiological data. We are grateful to the microbiology technicians in the participating laboratories for their contribution to the collection of the microbiological data.

### i-4-1-Health Study Group

Herman Goossens (University of Antwerp, Antwerp, Belgium and Antwerp University Hospital, Antwerp, Belgium, lead author, email: herman.goossens@uza.be), Lieke van Alphen (Maastricht University Medical Center +, Maastricht, the Netherlands), Nicole van den Braak (Avans University of Applied Sciences, Breda, the Netherlands), Caroline Broucke (Agency for Care and Health, Brussels, Belgium), Anton Buiting (Elisabeth-TweeSteden Ziekenhuis, Tilburg, the Netherlands), Liselotte Coorevits (Ghent University Hospital, Ghent, Belgium), Sara Dequeker (Department Care, Brussels, Belgium), Jeroen Dewulf (Ghent University, Ghent, Belgium), Wouter Dhaeze (Department Care, Brussels, Belgium), Bram Diederen (ZorgSaam Hospital, Terneuzen, the Netherlands), Helen Ewalts (GGD Hart voor Brabant, Tilburg, the Netherlands), Inge Gyssens (Hasselt University, Hasselt, Belgium), Casper den Heijer (GGD Zuid-Limburg, Heerlen, the Netherlands), Christian Hoebe (Maastricht University Medical Center+, Maastricht, the Netherlands and GGD Zuid-Limburg, Heerlen, the Netherlands), Casper Jamin (Maastricht University Medical Center+, Maastricht, the Netherlands), Patricia Jansingh (GGD Limburg Noord, Venlo, the Netherlands), Jan Kluytmans (Amphia Hospital, Breda, the Netherlands and University Medical Center Utrecht, Utrecht University, Utrecht, the Netherlands), Marjolein Kluytmans–van den Bergh (Amphia Hospital, Breda, the Netherlands and University Medical Center Utrecht, Utrecht University, Utrecht, the Netherlands), Stefanie van Koeveringe (Antwerp University Hospital, Antwerp, Belgium), Sien De Koster (University of Antwerp, Antwerp, Belgium), Christine Lammens (University of Antwerp, Antwerp, Belgium), Isabel Leroux (Ghent University Hospital,

Ghent, Belgium), Hanna Masson (Agency for Care and Health, Brussel, Belgium), Ellen Nieuwkoop (Elisabeth-TweeSteden Ziekenhuis, Tilburg, the Netherlands), Anita van Oosten (Admiraal de Ruyter Hospital, Goes, the Netherlands), Natascha Perales Selva (Antwerp University Hospital, Antwerp, Belgium), Merel Postma (Ghent University, Ghent, Belgium), Stijn Raven (GGD West-Brabant, Breda, the Netherlands), Paul Savelkoul (Maastricht University Medical Center+, Maastricht, the Netherlands), Annette Schuermans (University Hospitals Leuven, Leuven, Belgium), Nathalie Sleeckx (Proefbedrijf Pluimveehouderij VZW, Geel, Belgium), Krista van der Slikke (GGD Zeeland, Goes, the Netherlands), Arjan Stegeman (Utrecht University, Utrecht, the Netherlands), Tijs Tobias (Utrecht University, Utrecht, the Netherlands), Paulien Tolsma (GGD Brabant Zuid-Oost, Hertogenbosch, the Netherlands), Jacobien Veenemans (Admiraal de Ruyter Hospital, Goes, the Netherlands), Dewi van der Vegt (PAMM Laboratory for pathology and medical microbiology, Veldhoven, the Netherlands), Martine Verelst (University Hospitals Leuven, Leuven, Belgium), Carlo Verhulst (Amphia Hospital, Breda, the Netherlands), Pascal De Waegemaeker (Ghent University Hospital, Ghent, Belgium), Veronica Weterings (Amphia Hospital, Breda, the Netherlands and Radboud University Medical Center, Nijmegen, the Netherlands), Clementine Wijkmans (GGD Hart voor Brabant, Tilburg, the Netherlands), Patricia Willemse–Smits (Elkerliek Ziekenhuis, Geldrop, the Netherlands), Ina Willemsen (Amphia Hospital, Breda, the Netherlands).

## Author Contributions

**Conceptualization:** Jan Kluytmans, Herman Goossens.

**Data curation:** Sien De Koster, Natascha Perales Selva, Marjolein Kluytmans-Van den Bergh.

**Formal analysis:** Sien De Koster, Basil Britto Xavier, Samuel Coenen.

**Funding acquisition:** Jan Kluytmans, Herman Goossens.

**Investigation:** Sien De Koster, Natascha Perales Selva, Stefanie van Kleef-van Koeveringe, Wouter Dhaeze.

**Methodology:** Christine Lammens, Samuel Coenen, Youri Glupczynski, Isabel Leroux-Roels, Wouter Dhaeze, Christian J. P. A. Hoebe, Jeroen Dewulf, Arjan Stegeman, Jan Kluytmans.

**Project administration:** Christine Lammens, Marjolein Kluytmans-Van den Bergh, Jan Kluytmans, Herman Goossens.

**Supervision:** Basil Britto Xavier, Christine Lammens, Youri Glupczynski, Isabel Leroux-Roels, Wouter Dhaeze, Christian J. P. A. Hoebe, Jeroen Dewulf, Arjan Stegeman, Marjolein Kluytmans-Van den Bergh, Jan Kluytmans, Herman Goossens.

**Visualization:** Sien De Koster.

**Writing – original draft:** Sien De Koster.

**Writing – review & editing:** Basil Britto Xavier, Christine Lammens, Natascha Perales Selva, Stefanie van Kleef-van Koeveringe, Samuel Coenen, Youri Glupczynski, Isabel Leroux-Roels, Wouter Dhaeze, Christian J. P. A. Hoebe, Jeroen Dewulf, Arjan Stegeman, Marjolein Kluytmans-Van den Bergh, Jan Kluytmans, Herman Goossens.

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
