## [Decision Letter · Decision Letter 0]

29 Nov 2023

PONE-D-23-32355One Health surveillance of colistin-resistant Enterobacterales in Belgium and the Netherlands between 2017 and 2019PLOS ONE

Dear Dr. De Koster,

Thank you for submitting your manuscript to PLOS ONE. After careful consideration, we feel that it has merit but does not fully meet PLOS ONE’s publication criteria as it currently stands. Therefore, we invite you to submit a revised version of the manuscript that addresses the points raised during the review process.

We look forward to receiving your revised manuscript.

Kind regards,

Zhi Ruan, Ph.D.

Academic Editor

PLOS ONE

Journal Requirements:

   "The i-4-1-Health project was financed by the Interreg V Flanders-The Netherlands program, the cross-border cooperation program with financial support from the European Regional Development Fund (ERDF) (0215). Additional financial support was received from the Dutch Ministry of Health, Welfare and Sport (325911), the Dutch Ministry of Economic Affairs (DGNR-RRE/14191181), the Province of Noord-Brabant (PROJ-00715/PROJ-01018/PROJ-00758), the Belgian Department of Agriculture and Fisheries (no reference), the Province of Antwerp (1564470690117/1564470610014) and the Province of East-Flanders (E01/subsidie/VLNL/i-4-1-Health). The authors are free to publish the results from the project without interference from the funding bodies. FecalSwabs and tryptic soy broths were provided by Copan. The authors were free to publish the results from the project without interference by Copan."

5. One of the noted authors is a group or consortium "i-4-1-Health Study Group". In addition to naming the author group, please list the individual authors and affiliations within this group in the acknowledgments section of your manuscript. Please also indicate clearly a lead author for this group along with a contact email address.

Reviewers' comments:

Reviewer's Responses to Questions

**Comments to the Author**

1. Is the manuscript technically sound, and do the data support the conclusions?

Reviewer #1: Yes

Reviewer #2: Yes

2. Has the statistical analysis been performed appropriately and rigorously? 

Reviewer #1: Yes

Reviewer #2: Yes

3. Have the authors made all data underlying the findings in their manuscript fully available?

Reviewer #1: No

Reviewer #2: Yes

4. Is the manuscript presented in an intelligible fashion and written in standard English?

Reviewer #1: Yes

Reviewer #2: Yes

5. Review Comments to the Author

Reviewer #1: This manuscript explore colistin resistance bacteria and AMR genes in each one-health sector. This is interesting and attract the readers in the field. My minor comments appear below:

1. M & M: How the author selected the setting of the study? random or clustering or other?

2. M & M: Line 142-146, I think it will be better to move this paragraph to be ethical approval and inform consent.

3. M & M: How to choose farms? (line 155-157).

4. M 7 M: Line 167, please shortly described of Kluytmans-van der Bergh et al.

5. M & M: Line 173, What is antibiotics used in this study, please mention.

6. Result: What is genetic relationship of E. coli/Klebsiella/Enterobacter in this study? Do author perform core genome SNP?

7. Discussion: Line 562-563, Are they used colistin? I think it may be not, however, hospital stay is possible to get colistin resistance organism via oral-route.

8. In this study found mcr-9 and mcr-10 from human, that is different from several previous study which reported mcr-1 in healthy humans and patients. Especially, one report in Thailand showed mcr-1 carrying in carbapenem-resistant Enterobacterales. Please see https://www.frontiersin.org/articles/10.3389/fmicb.2020.586368/full and https://www.nature.com/articles/s41598-022-21836-7

This might be useful for the reader to discussion.

Reviewer #2: - Line 127, please write: ‘...in hospital patient, long term care facility (LTCF) residents and healthy.....’

- Line 133, the term ‘i-4-1-Health’ is not really known, so please provide one to two sentences to define.

- Line 363, write ‘mcr-5.1 sequences from Belgian pigs..’ also line 365 ‘mcr-9 and mcr-10- containing plasmids’. I mean that in all case the ‘mcr’ must be written with low letter even in the beginning of the sentence (not Mcr).

- Line 457, write ‘..atypical enteropathogenic E. coli (aEPEC, n=18 from all One Health sectors)......

- Line 466, you can use the term ‘high risk clonal lineages’ instead of ‘pandemic clones’.

- Line 468, write ‘K. pneumoniae ST15 (n=2), ST45 469 (n=7), ST101 (n=1), ST147 (n=1) and ST307 (n=1), and E. cloacae ST171 (n=1)..’

- Line 488, you wrote ‘Inter-host transmission between humans and livestock animals was not detected. However, clusters of related isolates were detected in all sectors indicating that transmission of ColR-E occurred in broiler and in pig farms, between children within the day care centres, and between patients residing in the LTCFs and the hospitals’. WHAT is the difference between ‘Inter-host transmission between humans and livestock animals was not detected’ and ‘However, clusters of related isolates were detected .... occurred in broiler and in pig farms, between children within the day care centres, and between patients residing in the LTCFs and the hospitals’. IT SEEMS that there is a contradiction. If you find same clone in different niches so this is ‘Inter-host transmission between humans and livestock animals’. You can explain this to me or you can verify or improve the sentences.

- Line 517, write ‘..However, colistin screening in humans is not included [17]’.

- Line 570, just a question, you wrote ‘Adenosine triphosphate (ATP) measurements also ...’ I am not specialized in this but what is the link between ATP and environmental contamination.

- I believe that mcr genes are mainly co-localized with ESBL genes, it is not clear if you find this linkage or not. Please highlight this phenomenon (if there is linkage or even there not) in the discussion section.

6. PLOS authors have the option to publish the peer review history of their article (what does this mean?). If published, this will include your full peer review and any attached files.

Reviewer #1: No

Reviewer #2: **Yes: **Mohamed Salah Abbassi

---

## [Author Response · Author response to Decision Letter 0]

4 Jan 2024

Dear Editor,

We thank the editor and reviewers for taking our manuscript into consideration for publication and for their constructive feedback. Hereby, we submit the revised version of our manuscript “One Health surveillance of colistin-resistant Enterobacterales in Belgium and the Netherlands between 2017 and 2019”. We addressed the editor’s and reviewer’s comments point by point. Please find the answers below.

Comments editor:

Answer: The style requirements were taken into account. 

Answer: The funding-related text was removed from the manuscript text.

 "The i-4-1-Health project was financed by the Interreg V Flanders-The Netherlands program, the cross-border cooperation program with financial support from the European Regional Development Fund (ERDF) (0215). Additional financial support was received from the Dutch Ministry of Health, Welfare and Sport (325911), the Dutch Ministry of Economic Affairs (DGNR-RRE/14191181), the Province of Noord-Brabant (PROJ-00715/PROJ-01018/PROJ-00758), the Belgian Department of Agriculture and Fisheries (no reference), the Province of Antwerp (1564470690117/1564470610014) and the Province of East-Flanders (E01/subsidie/VLNL/i-4-1-Health). The authors are free to publish the results from the project without interference from the funding bodies. FecalSwabs and tryptic soy broths were provided by Copan. The authors were free to publish the results from the project without interference by Copan."

Answer: The funders had no role, please adapt the online submission form with “The funders had no role in study design, data collection and analysis, decision to publish, or preparation of the manuscript”

Answer: The sequences were submitted to NCBI under bioproject PRJNA927131. This BioProject submission will be automatically released on 2024-02-02 or upon publication, whichever is first. The individual accession numbers for each isolate can be found in supplementary table 3. 

5. One of the noted authors is a group or consortium "i-4-1-Health Study Group". In addition to naming the author group, please list the individual authors and affiliations within this group in the acknowledgments section of your manuscript. Please also indicate clearly a lead author for this group along with a contact email address.

Answer: A list of individual authors and affiliations within the i-4-1-Health Study Group is provided in the Acknowledgements section of the manuscript. The information on the lead author for this group and contact email address was provided.

Answer: The ethics statement was moved to the Methods section of the manuscript. 

Comments reviewer 1:

Reviewer #1: This manuscript explore colistin resistance bacteria and AMR genes in each one-health sector. This is interesting and attract the readers in the field. My minor comments appear below:

1. M & M: How the author selected the setting of the study? random or clustering or other?

Answer: Sites were included based on the location. The study was conducted in the border region of Belgium (Flanders) and the Netherlands (South of the Netherlands). For the farms, an additional selection based on high antibiotic use according to the national benchmark value was performed. Samples were collected cross-sectionally. Within the hospitals, day care centres and long-term care facilities, persons that were willing to participate were included in the study. Within the farms, 30 samples were collected in a stratified-random sampling design. This information was included in the materials and methods section under ‘setting, study period and sample/strain collection’.

2. M & M: Line 142-146, I think it will be better to move this paragraph to be ethical approval and inform consent.

Answer: The details on how informed consent was obtained was moved to the Ethics Statement within the Materials and methods section.

3. M & M: How to choose farms? (line 155-157).

Answer: A more elaborate description of the farm selection is now provided in the materials and methods section under ‘setting, study period and sample/strain collection’.

4. M 7 M: Line 167, please shortly described of Kluytmans-van der Bergh et al.

Answer: This information is now added to the materials and methods section under ‘isolation of colistin-resistant Enterobacterales and antibiotic susceptibility testing’.

5. M & M: Line 173, What is antibiotics used in this study, please mention.

Answer: This information is now added in the materials and methods section under ‘isolation of colistin-resistant Enterobacterales and antibiotic susceptibility testing’.

6. Result: What is genetic relationship of E. coli/Klebsiella/Enterobacter in this study? Do author perform core genome SNP?

Answer: The genetic relationship of isolates was based on core genome multilocus sequence typing. This method was chosen because it is a powerful genotyping tool to delineate transmission routes. The data is shown in the minimum spanning trees in Fig 6. 

7. Discussion: Line 562-563, Are they used colistin? I think it may be not, however, hospital stay is possible to get colistin resistance organism via oral-route.

Answer: We did not collect data on colistin use in humans. Colistin is generally only used in hospitals and not in the general community or children. We agree with the reviewer that this might be confusing for the reader and removed this part. Indeed, hospital stay is a possible risk factor to acquire colistin-resistant organisms.

8. In this study found mcr-9 and mcr-10 from human, that is different from several previous study which reported mcr-1 in healthy humans and patients. Especially, one report in Thailand showed mcr-1 carrying in carbapenem-resistant Enterobacterales. Please see https://www.frontiersin.org/articles/10.3389/fmicb.2020.586368/full and https://www.nature.com/articles/s41598-022-21836-7

This might be useful for the reader to discussion.

Answer: The information was added to the discussion.

Comments reviewer 2:

- Line 127, please write: ‘...in hospital patient, long term care facility (LTCF) residents and healthy.....’

Answer: This was adapted in the manuscript text.

- Line 133, the term ‘i-4-1-Health’ is not really known, so please provide one to two sentences to define.

Answer: This information was added.

- Line 363, write ‘mcr-5.1 sequences from Belgian pigs..’ also line 365 ‘mcr-9 and mcr-10- containing plasmids’. I mean that in all case the ‘mcr’ must be written with low letter even in the beginning of the sentence (not Mcr).

Answer: This was adapted in the text.

- Line 457, write ‘..atypical enteropathogenic E. coli (aEPEC, n=18 from all One Health sectors)......

Answer: This was adapted in the text.

- Line 466, you can use the term ‘high risk clonal lineages’ instead of ‘pandemic clones’.

Answer: This was adapted in the text.

- Line 468, write ‘K. pneumoniae ST15 (n=2), ST45 469 (n=7), ST101 (n=1), ST147 (n=1) and ST307 (n=1), and E. cloacae ST171 (n=1)..’

Answer: This was adapted in the text.

- Line 488, you wrote ‘Inter-host transmission between humans and livestock animals was not detected. However, clusters of related isolates were detected in all sectors indicating that transmission of ColR-E occurred in broiler and in pig farms, between children within the day care centres, and between patients residing in the LTCFs and the hospitals’. WHAT is the difference between ‘Inter-host transmission between humans and livestock animals was not detected’ and ‘However, clusters of related isolates were detected .... occurred in broiler and in pig farms, between children within the day care centres, and between patients residing in the LTCFs and the hospitals’. IT SEEMS that there is a contradiction. If you find same clone in different niches so this is ‘Inter-host transmission between humans and livestock animals’. You can explain this to me or you can verify or improve the sentences.

Answer: We adapted the sentences to clarify. Transmission between humans and animals was not detected (inter-host transmission), however, animal-to-animal and human-to-human transmission was detected within sampling site and also between different sampling sites.

- Line 517, write ‘..However, colistin screening in humans is not included [17]’.

Answer: This was adapted in the text.

- Line 570, just a question, you wrote ‘Adenosine triphosphate (ATP) measurements also ...’ I am not specialized in this but what is the link between ATP and environmental contamination.

Answer: ATP measurements are an objective technique to measure biological contamination. ATP is a molecule present in all organic cells and can be measured using a luminometer. The technique is used to quantify the amount of organic matter (including bacterial contamination) on a surface in an objective and reproducible way. 

- I believe that mcr genes are mainly co-localized with ESBL genes, it is not clear if you find this linkage or not. Please highlight this phenomenon (if there is linkage or even there not) in the discussion section.

Answer: This linkage was not detected, however, the majority of isolates was sequenced using short-read sequencing which complicates this analysis. This was added to the discussion section.

The revisions that were recommended were answered and the manuscript was adapted accordingly. The style requirements were taken into account. Any funding-related text was removed from the manuscript. The role of funder in the financial disclosure can state “The funders had no role in study design, data collection and analysis, decisions to publish, or preparation of the manuscript”. The data availability statement can state “All data is included in this article or in supplementary information files. The sequences were submitted in NCBI under BioProject PRJNA927131.” This BioProject submission will be automatically released on 2024-02-02 or upon publication, whichever is first. The individual accession numbers for each isolate can be found in S3 Table. Thank you for your consideration of this revised manuscript.

Sien De Koster, PhD.

---

## [Decision Letter · Decision Letter 1]

18 Jan 2024

One Health surveillance of colistin-resistant Enterobacterales in Belgium and the Netherlands between 2017 and 2019

PONE-D-23-32355R1

Dear Dr. De Koster,

We’re pleased to inform you that your manuscript has been judged scientifically suitable for publication and will be formally accepted for publication once it meets all outstanding technical requirements.

Kind regards,

Zhi Ruan, Ph.D.

Academic Editor

PLOS ONE

Additional Editor Comments (optional):

Reviewers' comments:

Reviewer's Responses to Questions

**Comments to the Author**

1. If the authors have adequately addressed your comments raised in a previous round of review and you feel that this manuscript is now acceptable for publication, you may indicate that here to bypass the “Comments to the Author” section, enter your conflict of interest statement in the “Confidential to Editor” section, and submit your "Accept" recommendation.

Reviewer #1: All comments have been addressed

2. Is the manuscript technically sound, and do the data support the conclusions?

Reviewer #1: Yes

3. Has the statistical analysis been performed appropriately and rigorously? 

Reviewer #1: Yes

4. Have the authors made all data underlying the findings in their manuscript fully available?

Reviewer #1: Yes

5. Is the manuscript presented in an intelligible fashion and written in standard English?

Reviewer #1: Yes

6. Review Comments to the Author

Reviewer #1: (No Response)

7. PLOS authors have the option to publish the peer review history of their article (what does this mean?). If published, this will include your full peer review and any attached files.

Reviewer #1: No

---

## [Editor Report · Acceptance letter]

15 Feb 2024

PONE-D-23-32355R1 

PLOS ONE

Dear Dr. De Koster, 

I'm pleased to inform you that your manuscript has been deemed suitable for publication in PLOS ONE. Congratulations! Your manuscript is now being handed over to our production team.

Kind regards, 

on behalf of

Dr. Zhi Ruan 

Academic Editor

PLOS ONE